# The ecological drivers of variation in global language diversity

Xia Hua [1,2], Simon J. Greenhill[1,3], Marcel Cardillo[2], Hilde Schneemann[1,2,4] & Lindell Bromham[1,2]

Language diversity is distributed unevenly over the globe. Intriguingly, patterns of language diversity resemble biodiversity patterns, leading to suggestions that similar mechanisms may underlie both linguistic and biological diversification. Here we present the first global analysis of language diversity that compares the relative importance of two key ecological mechanisms – isolation and ecological risk – after correcting for spatial autocorrelation and phylogenetic non-independence. We find significant effects of climate on language diversity, consistent with the ecological risk hypothesis that areas of high year-round productivity lead to more languages by supporting human cultural groups with smaller distributions. Climate has a much stronger effect on language diversity than landscape features, such as altitudinal range and river density, which might contribute to isolation of cultural groups. The association between biodiversity and language diversity appears to be an incidental effect of their covariation with climate, rather than a causal link between the two.

[1] ARC Centre of Excellence for the Dynamics of Language, Australian National University, Canberra ACT 0200, Australia. [2] Macroevolution and Macroecology Group, Division of Ecology & Evolution, Research School of Biology, Australian National University, Canberra ACT 0200, Australia. [3] Max Planck Institute for the Science of Human History, Kahlaische Strasse 10, D-07743 Jena, Germany. [4] Meme Programme, University of Groningen, Nijenborgh 7, 9747 AG Groningen, The Netherlands. Correspondence and requests for materials should be addressed to X.H. (email: xia.hua@anu.edu.au)

The geographic distribution of the world's > 7500 languages is strikingly uneven[1] (Fig. 1). For example, Papua New Guinea represents over 10% of the world's languages in < 0.5% of the world's land area[1]. In contrast, the Russian Federation covers 11% of the world land area but accounts for only 1.5% of the world's languages. The highly uneven distribution of languages remains a major unsolved problem in linguistics[2,3]. Yet, there are broad geographic patterns in the distribution of languages, suggesting a strong role for environmental determination of language diversity. The most notable of these patterns are latitudinal gradients: language diversity increases towards the equator[4–11], and languages in the tropics tend to be restricted to smaller areas than languages at higher latitudes[4,6,8].

Two broad kinds of ecological mechanism have been offered to explain geographic variation in language diversity: isolation and ecological risk. Isolation mechanisms are associated with landscape and geographic features that act as barriers to human movement. Such physical barriers may reduce interaction between groups, slowing the spread of linguistic variants among neighboring populations, leading to the accumulation of language changes that distinguish each language from its neighbors[12]. Previous studies have suggested that geographic correlates of language diversity, such as river density[13], landscape roughness[13,14], elevation range[15], and habitat diversity[14], point to a role for isolation in generating language diversity.

The ecological risk mechanism for language diversity predicts an association between language diversity and climatic factors such as seasonal temperature variation and yearly rainfall[16–19]. Smaller social groups are presumed to be more likely to be stable and self-sufficient in areas with a more abundant and reliable year-round food supply. In contrast, areas of high seasonality or unpredictable rainfall may require communities to form social bonds across larger regions to obtain food and resources when they are scarce. In support of the ecological risk hypothesis, various studies have shown correlations between language diversity and environmental productivity[20], mean growing season[16–18], rainfall[4,20,21], and temperature[4].

Areas of low ecological risk tend to occur in high-productivity environments that often also promote high plant and animal diversity[22–24]. Indeed, countries with high vertebrate and flowering plant diversity tend to have high language diversity[25], and there are significant correlations between biological and language diversity at regional[20,26] and global scales[27–29]. It has also been suggested that high biodiversity could have a direct link with language diversity, if small social groups possess specialized knowledge of local biodiversity[26,30], or if group boundaries are maintained to actively control local biodiversity resources[31].

Although previous studies have reported correlations between language diversity and a number of environmental or landscape variables, there has yet to be a comprehensive global analysis. Many of these variables co-vary and tend to be clustered in space and more similar between related languages[32–34]. Here, we present a global-scale analysis of language diversity that comprehensively deals with these statistical complexities. Our aims are to untangle and clarify the large-scale patterns of association between language diversity and key environmental variables, and thereby determine the relative strength of support for the alternative ecological mechanisms that may drive global patterns of language diversity.

In doing so, we provide the first analysis to explicitly weigh the relative contribution of the key proposed ecological mechanisms for language diversification—climate, landscape, and biodiversity —while controlling for key statistical challenges arising from the relatedness of languages and their non-random spatial patterns, and covariation amongst variables. We show that the influence of climate is consistent with predictions that longer growing seasons allow greater diversity of languages per area[16–19]. Because we use a larger selection of climatic features than previous analyses, we are able to show that seasonality of temperature and precipitation provides additional explanation for patterns of language diversity, suggesting that the year-round predictability of conditions for growth is likely to be a key driver of language diversity in an area.

We find less support for the hypothesis that landscape features stimulate language diversity by limiting human movement and dividing populations into smaller speaker groups. We also find no evidence supporting a direct link between biological diversity and language diversity[25,26,30,31], showing that this association is more likely owing to covariation of both biodiversity and language diversity with climatic factors and landscape features. However, although environmental factors show a significant influence on language diversity, we still find regions with greater variation in language diversity than can be explained by climate, landscape, or biodiversity, highlighting a role for other processes in shaping language diversity[5,35].

## Results

**Autocorrelation in language diversity**. To weigh the relative explanatory power of the isolation and ecological risk hypotheses,

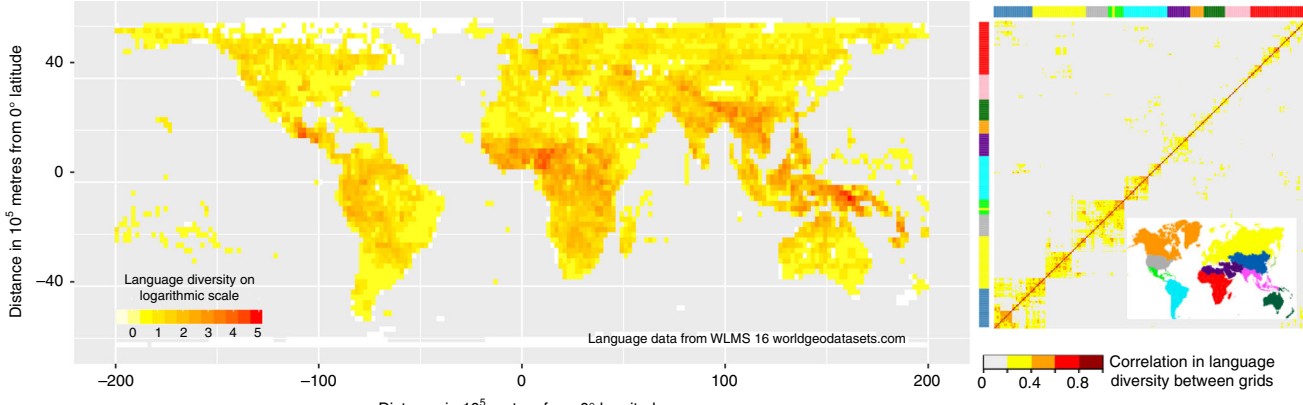

**Fig. 1** Global distribution of language diversity. Left panel: values on a logarithmic scale of number of languages are shown for 200 × 200 km cells of an equal-area grid. Right panel: correlation in language diversity between each pair of grid cells owing to spatial autocorrelation and phylogenetic relatedness. Correlation coefficient is estimated from our generalized least squares model that includes all the climatic and landscape variables as predictors (see Methods). Correlated grid cells are roughly clustered into nine geographic regions, so we color code the rows and columns by these regions. Grid cells within East Asia, Europe, and the Americas are more autocorrelated than grid cells within the other regions

we tested the association between language diversity and six climatic variables, four geographic variables, two human population variables and four biodiversity measures. Our analyses were based on global-scale datasets of the geographic distribution of 6425 languages[1], high-resolution climatic and geographic data layers, and global biodiversity datasets. We used the number of languages whose distribution overlaps each cell of a global equal-area grid as the measure of language diversity. A grid-based approach eliminates variation in language diversity and other variables due to differences in land area. It also allowed us to repeat analyses under three different spatial resolutions, as previous studies have shown that spatial resolution can influence tests for latitudinal gradients in language diversity[7,26].

In order to untangle causal connections from incidental associations, we need to account for sources of covariation in our data. In particular, we need to address spatial autocorrelation and phylogenetic non-independence[32–34]. Grid cells that are located near each other are likely to have similar values of climatic and landscape variables, contain related human cultures and languages, and share much of their flora and fauna. If there are any particular features of cultures or languages that correlate with language diversity, then they will tend to co-vary with environmental features and biodiversity measures, whether or not there is any causal connection between them.

Our analyses show that regions in close geographic proximity, and with a high degree of language relatedness, tend to be more similar in their language diversity (Fig. 1 and Supplementary Figure 1). Spatial autocorrelation and phylogenetic relatedness account for 15% of the variance in language diversity in analyses at low spatial resolution, 18% at medium resolution and 5% at high resolution, even after removing highly correlated grid cells (see Methods). This confirms the need to account for spatial proximity and phylogenetic relatedness between grid cells when testing for correlates of language diversity.

After correcting for spatial autocorrelation and phylogenetic relatedness, there is a latitudinal gradient in language diversity, with more languages near the equator than at higher latitudes (Fig. 1). The regression coefficient of absolute latitude against language diversity is significantly negative under all resolutions (low: $t = -2.58$; $p = 0.011$; medium: $t = -2.49$; $p = 0.014$; high: $t = -5.22$; $p < 0.001$).

**Climatic effect on language diversity**. We tested six climatic variables for associations with language diversity: mean annual temperature, mean annual precipitation, temperature seasonality, precipitation seasonality, net primary productivity, and mean annual growing season (Supplementary Figure 2). Among these climatic variables, precipitation seasonality has the strongest association with language diversity in low resolution analyses, and temperature seasonality has the strongest association with

language diversity in medium and high-resolution analyses, independently of their covariation with other climatic variables (Table 1). The six climatic variables also provide sufficient explanation for the latitudinal gradient in language diversity, because adding latitude as an explanatory variable in the model in addition to the six climatic variables does not significantly increase the model fit under any resolution (low: $LR = 0.94$, $p = 0.33$; medium: $LR = 1.15$, $p = 0.28$; high: $LR = 2.03$, $p = 0.15$).

The ecological risk hypothesis may provide an explanation for the association between language diversity and seasonality, which predicts higher language diversity in regions with longer periods of reliable food production, by allowing smaller cultural groups to be self-sufficient[16–19]. This hypothesis makes testable predictions about the associations between climate, population size and density, and language diversity. We followed previous studies[16–19] in using mean growing season (the number of days per year suitable for growing crops) as an indicator of ecological risk, although our results indicate that temperature seasonality may be a better predictor of the influence of environment on language diversity. The ecological risk hypothesis predicts that longer growing seasons will result in reduced area per language and smaller speaker population sizes. We find evidence to support both of these predictions. Longer growing seasons are associated with a greater number of languages per grid cell at all three resolutions (Table 2), consistent with a reduction in range sizes of languages allowing tighter packing of languages (as language polygons are largely non-overlapping, smaller average range size allows more languages fit into a given area). The increase in language diversity is not simply a result of areas with long growing seasons supporting a greater number of people, because mean growing season has a significant positive association with language diversity beyond its covariation with population density (Table 2). Mean growing season is negatively associated with minimum speaker population size (the population size of the smallest language in a grid cell) under medium and high resolutions (Table 2), consistent with the prediction that smaller cultural groups are more able to persist in areas of longer growing season. There is no association between mean growing season and the average speaker population size of all the languages in a grid cell, so increased packing is primarily a result of reduction in language range size in areas with longer growing seasons, rather than being attributable to a reduction in the average size of cultural groups within those high-diversity areas.

Our results are broadly consistent with the ecological risk hypothesis, because mean growing season is associated with both the minimum group size and the number of languages per grid cell. However, we find that seasonality in temperature and precipitation have additional associations with language diversity that is not attributable to mean growing season. This is consistent with a recent study that supports associations between language diversity and average amount of precipitation in the wettest

**Table 1 Climatic effects on language diversity**

| Predictor | Low (n = 216) | Medium (n = 192) | High (n = 366) |
|---|---|---|---|
| Annual mean precipitation | 0.56 (0.577) *0.32* | 1.32 (0.190) *1.67* | 1.74 (0.083) *1.27* |
| Annual mean temperature | 0.04 (0.968) *0.00* | −0.90 (0.369) *0.69* | 1.15 (0.251) 0.80 |
| Precipitation seasonality | **2.15 (0.032)** *4.54* | 1.54 (0.126) *2.01* | 0.10 (0.920) 0.55 |
| Temperature seasonality | −1.14 (0.257) *1.30* | **−2.43 (0.016)** *4.76* | **−2.24 (0.026)** *4.12* |
| Net primary productivity | 1.86 (0.064) *3.54* | 1.65 (0.101) *2.69* | 1.58 (0.114) *3.38* |
| Mean annual growing season | 1.29 (0.199) *1.72* | 1.06 (0.290) *1.16* | 0.85 (0.395) *0.20* |

We list the *t* value and the *p* value (in parentheses) of each predictor in a generalized least squares regression that includes all the six eco-climatic predictors (*n* is the number of grid cells used in the analysis at low, medium, or high resolution). Two additional parameters are the intercept and the coefficient for land coverage. Because collinearity can inflate the standard error of regression coefficient, we also conduct likelihood ratio (*LR*) tests to assess if adding a predictor significantly increases model fit. If so, the predictor has a significant effect on language diversity beyond its covariation with other predictors. Significant results are in bold. *LR* value is shown in italic, after *t* value (*p* value)

**Table 2 Predictions of the ecological risk hypothesis**

| Response | Predictor | t | p | Does mean growing season show significant association with response beyond its covariation with predictor? |
|---|---|---|---|---|
| **Low resolution (n = 216)** | | | | |
| Language diversity | Mean growing season | **3.96** | **<0.001** | NA |
| Average population size | Mean growing season | −1.02 | 0.309 | NA |
| Min. population size | Mean growing season | −1.72 | 0.087 | NA |
| Population density | Mean growing season | **3.22** | **0.002** | NA |
| Language diversity | Latitude | **−2.58** | **0.011** | Yes: LR = 14.00, p < 0.001 |
| Language diversity | Population density | −0.59 | 0.556 | Yes: LR = 15.85, p < 0.001 |
| **Medium resolution (n = 192)** | | | | |
| Language diversity | Mean growing season | **4.70** | **<0.001** | NA |
| Average population size | Mean growing season | −1.92 | 0.056 | NA |
| Min. population size | Mean growing season | **−3.31** | **0.001** | NA |
| Population density | Mean growing season | **3.97** | **<0.001** | NA |
| Language diversity | Latitude | **−2.49** | **0.014** | Yes: LR = 15.38, p < 0.001 |
| Language diversity | Population density | −0.30 | 0.763 | Yes: LR = 18.53, p < 0.001 |
| **High resolution (n = 366)** | | | | |
| Language diversity | Mean growing season | **5.57** | **<0.001** | NA |
| Average population size | Mean growing season | −1.83 | 0.068 | NA |
| Min. population size | Mean growing season | **−3.13** | **0.002** | NA |
| Population density | Mean growing season | **4.83** | **<0.001** | NA |
| Language diversity | Latitude | **−5.22** | **<0.001** | Yes: LR = 29.68, p < 0.001 |
| Language diversity | Population density | 0.66 | 0.512 | Yes: LR = 29.07, p < 0.001 |

We list the $t$ value and the $p$ value of the predictor in each generalized least squares regression for the response variable ($n$ is the number of grid cells used in the analysis at low, medium, or high resolution). Each model includes three parameters: intercept, coefficient of land coverage, and coefficient of the predictor. Significant results are in bold. To test if mean growing season shows significant association with the response variable beyond its covariation with the predictor, we conduct a likelihood ratio test on whether adding mean growing season as an additional predictor significantly increase the model fit

quarter and temperature in the warmest quarter[36]. Although growing season is defined by the number of days above a specific minimum temperature and moisture availability, seasonality will reflect both minimum and maximum of temperatures and moisture. Therefore, this result may suggest that climatic extremes across seasons shape language diversity, in addition to average length of growing season.

**Landscape effect on language diversity**. To examine the effect of isolation on language diversity, we tested four landscape variables that have been suggested to influence patterns of human movement and therefore contribute to the isolation of cultural groups: mean altitude, altitudinal range, landscape roughness, and river density (Supplementary Figure 2). Higher river density is associated with greater language diversity at low and medium resolutions, beyond its covariation with climatic variables and the other landscape variables (Table 3). Although this result is consistent with previous proposals that rivers act to isolate populations into smaller language groups[13], we find little additional support for this hypothesis. Although river density is associated with smaller minimum speaker population size at medium resolution (Table 3), there is no association between river density and average speaker population size (controlling for the effects of population density). These observations suggest that the association between river density and language diversity is more akin to the ecological risk hypothesis than to the isolation hypothesis, because rivers seem to allow the persistence of smaller speaker populations, but not to divide human populations into smaller speaker populations. In this sense, rivers may act more as an ecological resource than a barrier to interaction.

Similarly, although altitudinal range is associated with language diversity at high resolution with marginal significance, there is no evidence that this is caused by isolation, as altitudinal range does not result in reduction in speaker population size, even when

controlling for population density (Table 3). Although landscape roughness is significantly associated with language diversity when altitudinal range is not included in the model ($t = 2.87$; $p = 0.004$), we find no significant association between landscape roughness and language diversity beyond its covariation with climatic variables and the other landscape variables under the three resolutions, and no statistically significant negative association between landscape roughness and speaker population size (Table 3).

In contrast to a previous study that described river density and landscape roughness as universal determinants of language diversity[13], we find little evidence that landscape variables have a strong or consistent influence on language diversity. Although we use similar data to Axelson & Manrubia[13], there are a number of differences in our analytical approach. To compare our results to theirs, we reanalyze our data using their method, fitting continent-specific parameter values and not including altitudinal range. Without correcting for spatial and phylogenetic non-independence among grid cells, we get similar results to Axelson & Manrubia[13], namely that river density and landscape roughness have significant associations with language diversity in most continents (Supplementary Table 1). But when we correct the data for non-independence among grid cells, neither river density nor landscape roughness has a significant association with language diversity in any continent (Supplementary Table 1). We therefore conclude that the previous result was driven primarily by spatial autocorrelation and phylogenetic non-independence, with the similarity in both landscape variables and language diversity between neighboring grid cells generating spurious correlations.

In conclusion, we find little consistent support for effect of landscape factors on language diversity. Although we find associations between language diversity and river density, altitudinal range and landscape roughness, these landscape

**Table 3 Landscape effects on language diversity and speaker population size**

| Response | Predictor | Low (n = 216) | Medium (n = 192) | High (n = 366) |
|---|---|---|---|---|
| Language diversity | Average altitude | −1.21 (0.228) *1.16* | 0.70 (0.483) *0.52* | 0.21 (0.830) *0.92* |
| | Altitudinal range | 1.67 (0.096) *2.93* | 1.01 (0.312) *1.09* | **1.98 (0.049)** ***4.03*** |
| | Landscape roughness | 0.13 (0.900) *0.02* | −0.00 (0.999) *0.00* | 0.98 (0.328) *0.99* |
| | River density | **3.02 (0.003)** ***9.22*** | **2.44 (0.016)** ***6.21*** | 1.42 (0.157) *2.07* |
| Average speaker population size | Average altitude | 0.44 (0.662) *0.20* | 0.74 (0.463) *0.56* | 0.73 (0.463) *0.55* |
| | Altitudinal range | 1.21 (0.229) *1.44* | −0.65 (0.515) *0.44* | −1.20 (0.232) *1.45* |
| | Landscape roughness | −0.82 (0.412) *0.70* | 1.24 (0.215) *1.59* | 1.94 (0.053) *3.76* |
| | River density | −1.02 (0.310) *1.06* | −1.20 (0.234) *1.44* | −0.47 (0.641) *0.21* |
| Minimum speaker population size | Average altitude | 0.50 (0.616) *0.27* | 0.81 (0.420) *0.62* | 1.20 (0.232) *1.46* |
| | Altitudinal range | 0.22 (0.824) *0.05* | −1.19 (0.238) *1.22* | −1.74 (0.083) *3.06* |
| | Landscape roughness | −1.25 (0.214) *1.64* | 1.57 (0.118) *2.35* | 1.04 (0.301) *1.08* |
| | River density | −0.43 (0.668) *0.20* | **−2.24 (0.026)** ***5.04*** | −1.32 (0.187) *1.68* |

We list the *t* value and the *p* value (in parentheses) of each landscape variable in each generalized least squares regression (*n* is the number of grid cells used in the analysis at low, medium, or high resolution). Models with language diversity include all the six climatic and four landscape variables. Models with population size includes all the four landscape variables and population density. Two additional parameters are the intercept and the coefficient for land coverage. We also conduct likelihood ratio test to test if adding a landscape variable significantly increases model fit. If so, the variable has a significant effect on language diversity beyond its covariation with climatic variables and the other landscape variables. Significant results are in bold. *LR* value is shown in italic, after *t* value (*p* value)

**Table 4 Association between biodiversity and language diversity**

| Biodiversity | Low (n = 216) | Medium (n = 192) | High (n = 366) |
|---|---|---|---|
| Plant diversity | 1.32 (0.188) *1.78* | 0.76 (0.449) *0.60* | 0.28 (0.783) *0.10* |
| Amphibian diversity | 0.38 (0.705) *0.14* | −0.67 (0.507) *0.41* | −0.30 (0.761) *0.10* |
| Mammal diversity | **3.12 (0.002)** ***9.09*** | 1.79 (0.075) *3.00* | **2.23 (0.027)** ***11.32*** |
| Bird diversity | **3.74 (<0.001)** ***12.23*** | **2.66 (0.009)** ***6.65*** | **2.99 (0.003)** ***15.37*** |

We list the *t* value and the *p* value (in parentheses) of a biodiversity variable in a generalized least squares regression that includes the biodiversity variable and all the six climatic and four landscape variables (*n* is the number of grid cells used in the analysis at low, medium, or high resolution). Two additional parameters are the intercept and the coefficient for land coverage. We also conduct likelihood ratio (LR) test to test if adding the biodiversity variable significantly increases model fit. Significant results are in bold. *LR* value is shown in italic, after *t* value (*p* value)

factors have much less influence on language diversity than climatic factors, and there is little indication that this is caused by the division of human populations into smaller, isolated cultural groups. Instead, previous results suggesting river density and landscape roughness are universal determinants of language diversity[13] may have been driven by autocorrelation among grid cells.

**Link between language diversity and biodiversity**. We now ask if biodiversity provides any additional explanation for language variation beyond covariation with climate and landscape factors. Adding mammal or bird diversity as additional predictors to the climatic and landscape variables significantly improves model fit, but adding vascular plant and amphibian diversity do not provide additional explanatory power (Table 4). Adding biome to the analysis increases model fit above climate variables at low resolution, suggesting that ecosystem structures may influence language diversity, however it does not provide significant explanatory power above the effect of climate at medium and high resolutions (low: *LR* = 27.01, *p* = 0.02; medium: *LR* = 14.91, *p* = 0.38; high: *LR* = 11.83, *p* = 0.62).

Why are bird and mammal diversity associated with language diversity? There is no evidence that this is due to a direct causal relationship between biodiversity and language diversity, because there is no consistent relationship between these biodiversity measures and residual variation in language diversity, above and beyond that explained by climate and landscape (Supplementary Table 2). Instead, the increase in model fit when bird and mammal diversity are added to the model of language diversity, climate and landscape, seems to be driven primarily by regions that have both low language diversity and low species diversity, particularly the Sahara, the Arabian Peninsula, and the Tibetan

Plateau (Fig. 2 and Supplementary Figure 3), which present harsh environmental conditions for birds and mammals (including humans). These are not the only regions of low diversity but they seem to have a disproportionate influence on the relationship between mammal and bird diversity and language diversity (Supplementary Figure 4). Running the high-resolution analysis without these low diversity areas, we find that adding mammal or bird diversity as additional predictors to the climatic and landscape variables no longer increases model fit (*n* = 334, mammal: *LR* = 1.92, *p* = 0.17; bird: *LR* = 3.67, *p* = 0.07), although this reduced data still show results for the climatic and landscape effects that are similar to those from the complete data set. Even when these low diversity areas are excluded from the analysis, temperature seasonality remains the strongest predictor for language diversity of all of the climatic variables (*t* = −2.34, *p* = 0.02) and altitudinal range remains the strongest predictor of language diversity of the landscape variables (*t* = 2.27, *p* = 0.02). These results suggest that the low diversity areas have a significant effect on the association between biodiversity and language diversity, but they are not responsible for the broader association between language diversity and climatic and landscape effects.

In conclusion, we find that the association between language diversity and biodiversity appears to be largely a result of their covariation with common climatic and landscape factors, and any additional increase in model fit between language diversity and mammal and bird diversity is likely due to the disproportionate effect of a few regions of harsh environment that reduce both biodiversity and language diversity.

**Residual variation in language diversity**. The six climatic variables and the four landscape variables together explain 45% of the

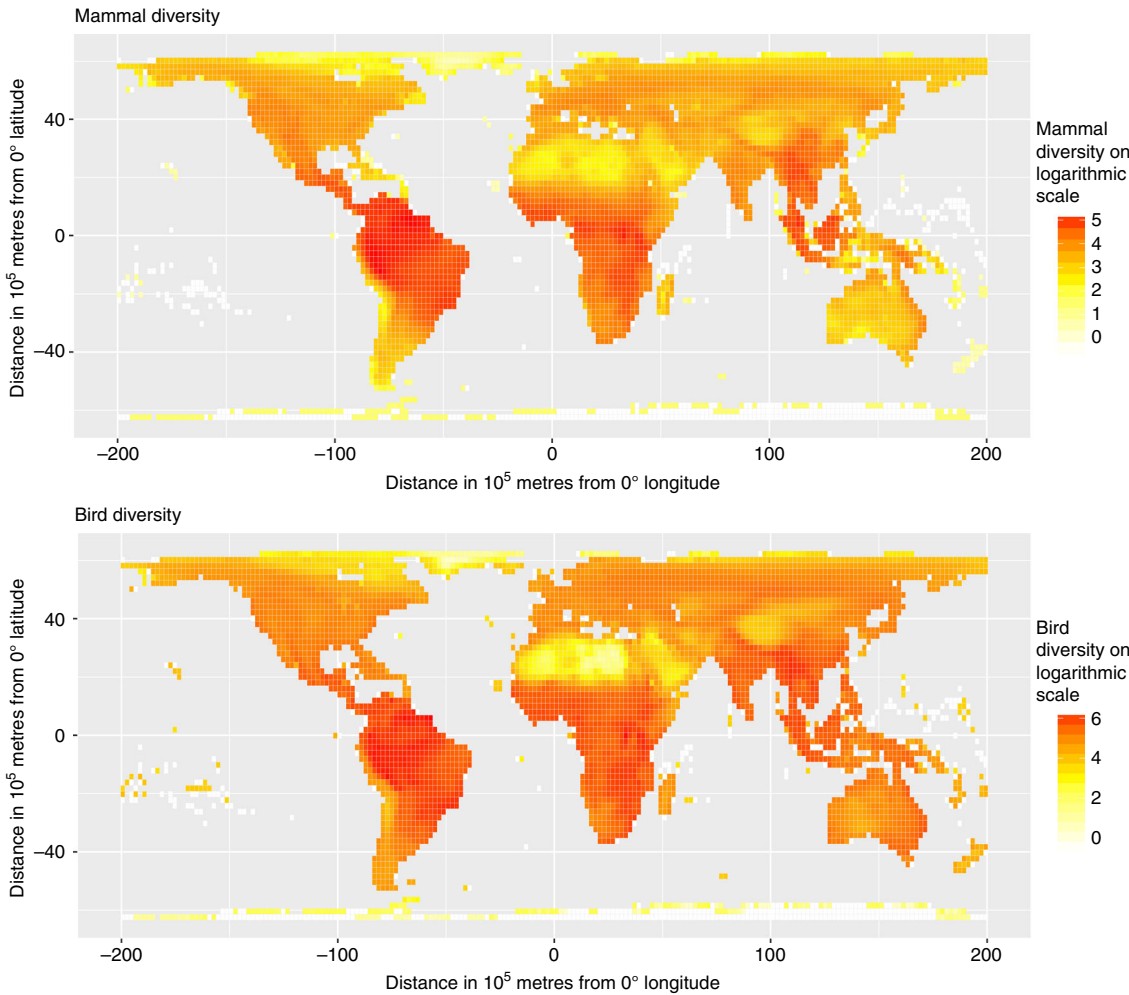

**Fig. 2** Global distribution of mammal diversity and bird diversity. Values on logarithm scale of number of species are shown for 200 × 200 km cells of an equal-area grid. For amphibian and plant diversity see Supplementary Figure 3

variance in language diversity under low resolution, 31% under medium resolution, and 27% under high resolution (after correction for phylogenetic and spatial non-independence). About 80% of this explanatory power is contributed by the six climatic variables under the three resolutions. Measures of biodiversity do not appear to add additional explanatory power beyond their covariation with climatic factors, above and beyond the influence of several key areas of low diversity.

What accounts for the remaining variation in language diversity? Fig. 3 shows the distribution of the residuals in language diversity after removing the climatic and landscape effects on language diversity. We can identify areas of high unexplained language diversity as the red grid cells with residuals ≥ 1.96 standard deviations higher than predicted by the climatic and landscape variables alone. These grid cells are concentrated in four regions—New Guinea, Eastern Himalaya, West Africa, and Mesoamerica. Language diversity in grid cells with residuals ≤ −1.96 (blue) is lower than we would predict based on the climatic and landscape variables, most notably in the lower Amazon Basin of South America.

There are several possible explanations for these areas of relative excess or paucity of languages, beyond that predicted by the climate and landscape variable. One is that they reflect relative completeness of language documentation. For example, Amazonia is considered an area of high language diversity[27], but incomplete documentation in the central areas of this region have led to it being described as the least known and least understood

linguistic region[37]. Therefore, the true number of languages may be higher than the documented number of languages. However, it seems unlikely that the opposite effect (over-reporting of language diversity) would explain the areas of high unexplained language diversity.

Alternatively, it may be that other factors contribute significantly to shaping language diversity that are not captured by climate variables (representing the ecological risk hypothesis) nor by landscape variables (representing isolation mechanisms). For example, regions of higher than expected language diversity may have had a longer period of in situ language diversification, or have undergone a higher rate of diversification, leading to a greater accumulation of languages in these regions than in other regions of similar climate. One way to investigate the influence of time or diversification rate on diversity is to use a phylogeny that contains information on the relative timing of diversification events in order to compare the timescale and rate of diversification in different regions[24,38]. Although phylogenies are available for the languages within some language families[39–44], and a global distance-based phylogeny[45], there is currently no dated phylogeny of the world's languages, nor is there general agreement on the relationships or age of language families. Therefore, we lack the means to make a quantitative comparison of duration or rates of diversification between the majority of grid cells (those that contain languages from different families or languages not contained in comprehensive phylogenies).

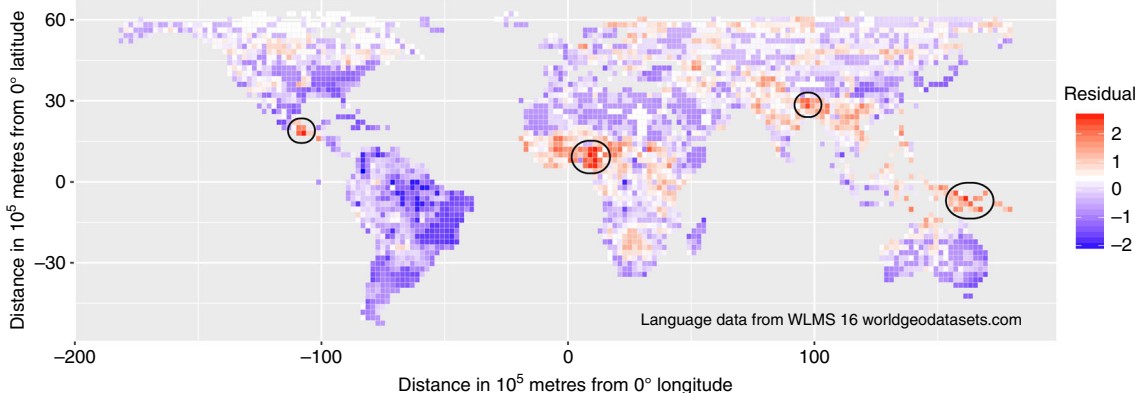

**Fig. 3** Global distribution of residuals in language diversity. Residuals after accounting for the climatic and landscape effects on language diversity are shown for 200 × 200 km grid cells of an equal-area grid. Aggregations of grid cells with residuals ≥ 1.96 (red) are circled. These indicate four regions of higher than expected language diversity, compared with regions of similar climate and landscape (New Guinea, Eastern Himalaya, West Africa, and Mesoamerica). Areas of lower than expected language diversity with residuals ≤ −1.96 (blue) are distributed in South America, mostly in the Amazon basin. The figure only shows grid cells for which we have relevant data

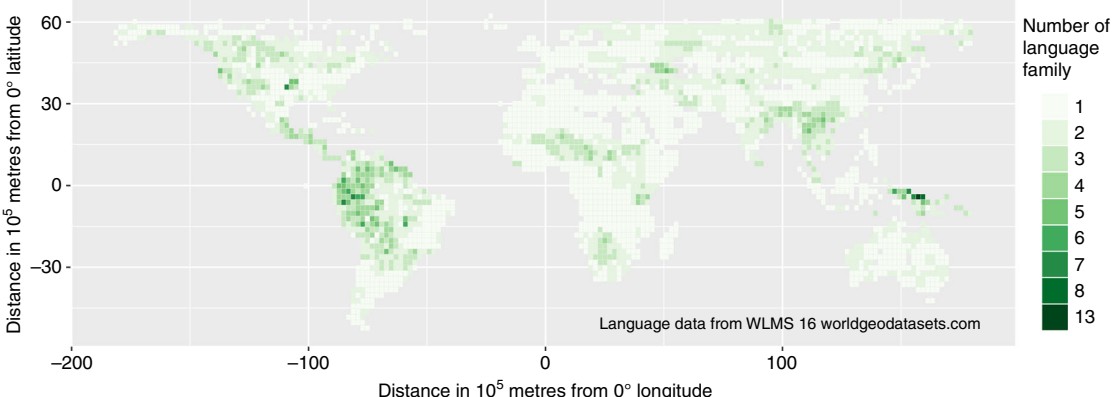

**Fig. 4** Global distribution of the number of language families. Numbers of language families are shown for 200 × 200 km cells of an equal-area grid. Language family is defined by the World Language Mapping System taxonomy[1]. Language isolates are treated as distinct families. Number of language families within a grid cell is calculated as the number of language families that include at least one language distributed in the grid cell. The figure only shows grid cells for which we have relevant data

Nevertheless, we can make a qualitative comparison of the relative depth of divergence represented in each grid cell if we make the simple assumption that languages from the same language family diverged more recently than languages from different families. Number of language families per grid cell is a significant predictor of residuals in language diversity under the three resolutions (low: $t = 4.65$, $p = < 0.001$; medium: $t = 6.27$, $p = < 0.001$; high: $t = 8.83$, $p = < 0.001$; Supplementary Figure 5). However, we are hesitant to draw strong conclusions from this pattern. For example, although New Guinea has more language families per grid cell than most other regions, the other areas of high unexplained language diversity do not have unusually high language family diversity, and some areas with many language families do not have high language richness (Fig. 4; Supplementary Figure 5). Clearly, this is not an ideal analysis of variation in time for diversification, as we cannot standardize time or rate of language evolution across families without a global dated phylogeny. But it suggests that time to diversification may be a profitable area of enquiry once complete language phylogenies become available.

In addition to factors relating to data completeness and time for diversification, we expect a large number of other factors to have influenced patterns of language diversity, which were not included in this study owing to our focus on the influence of

climate, landscape, and biodiversity. Some of these additional factors may have global patterns of influence. For example, it has been suggested that the relative explanatory power of climate on language diversity is stronger for foraging and pastoral societies, and less so for agricultural societies[5,35]. Subsistence strategy is also strongly influenced by climatic factors[46]. The areas of greater language diversity than would be predicted using environmental factors alone predominantly coincide with areas where the dominant subsistence strategy is plant-based agriculture[46]. However, it is unlikely that this provides a strong explanation for these hotspots because there are many more regions dominated by agriculture that do not have higher than expected language diversity.

It is also important to acknowledge that our analysis is based on a contemporary snapshot of language diversity, and uses only current climate information, therefore, we are unable to capture the influence of past environmental variation. Nor can we account for the influence of changing patterns of cultural diversity, political complexity, or subsistence patterns over time or space[35]. The identification of environmental factors associated with patterns of language distribution and diversity does not deny the role of historically contingent events unique to each culture. Human history is influenced by a great diversity of factors, including conflict, political structures, and patterns of human

migration. But, on top of these influences, we detect consistent influences of environmental factors that add to the millieu of factors impacting on patterns of human diversity.

The overall picture supported by our analyses is that environmental factors are a significant determinant of global variation in the diversity of human languages, as they are for global variation in biodiversity. Associations between global patterns of language diversity and climate are consistent with the ecological risk hypothesis, that stable productive climates allow human cultures to persist in smaller, more localized groups. Our results offer less support for isolation mechanisms as global drivers of language diversity. Although there are significant associations between language diversity and river density, altitudinal extent and landscape roughness, landscape factors have less explanatory power than climate, and the patterns are not indicative of an mechanism that divides human populations into smaller, isolated cultural groups. The association between biodiversity and language diversity is likely owing to an incidental association between language and species richness driven by shared causal factors such as climate and landscape. The importance of macroevolutionary influences such as time to accumulate diversity or the rate of language diversification are yet to be explored in detail.

## Methods

**Data collection**. All analyses are based on grid cells for a global equal-area projection at three different resolutions: low resolution with grid cell size $1000 \times 1000$ km, medium resolution with grid cell size $500 \times 500$ km, and high resolution with grid cell size $200 \times 200$ km. We excluded grid cells where no language distributions overlap that cell.

Language distributions were compiled from the World Language Mapping System v16 (http://worldgeodatasets.com) that includes information on the geographic distribution of 6425 languages[1]. These distribution data represent the likely spatial range of the traditional linguistic homelands of languages, and do not include in a language polygon speakers in other areas, such as migrant populations, nor do they represent official languages defined by political boundaries.

Language diversity was calculated by overlaying the language range polygons with the global grid using the R packages "sp" and "raster"[47–49] and counting the number of languages whose distribution overlaps all or part of a grid cell. We included the percentage of a grid cell covered by land as an independent variable in each regression model. For islands that cover < 1% of the area of the grid cell, land coverage was set to 0.01 unless the exact number could be derived from the language data.

Both the ecological risk and isolation hypotheses make predictions about the relationship between speaker population size and climate or landscape factors. We included both the minimum and the average size of speaker populations of all the languages present in each grid cell, based on current available estimates of the number of native speakers of a language resident in a country as recorded in WLMS (the number of L1 speakers in the country that spans a grid cell). These data do not capture change in number of speakers over time or historical changes in the geographic distribution of L1 speakers, but instead represent a snapshot of current speaker population distribution, which should provide a reflection of general patterns associated with population differences between languages[50]. To control for regional variations in number of people per grid cell, we used total human population density from the Gridded Population of the World database[51].

We included four climatic variables in our analysis: annual temperature, annual precipitation, temperature seasonality, and precipitation seasonality, averaged over each grid cell. We also included two variables derived from eco-climatic factors: net primary productivity and mean growing season of a grid cell. Net primary productivity data were derived from the Socioeconomic Data and Applications Center[52]. Data on growing season were obtained from the Global Agro-ecological Zones Data Portal version v3.0[53], which is calculated as the number of days per year suitable for growing crops based on precipitation, evapotranspiration and soil moisture holding capacity. The other climatic variables were obtained from the Worldclim global climate data set v1.4[54].

We included four variables describing landscape factors that have been previously suggested to influence population movement and range expansion: average altitude, altitudinal range, landscape roughness, and river density in each grid cell. Altitude data were obtained from Worldclim[54]. Landscape roughness data were calculated as the autocorrelation in altitude[13] (at every 1 km along 100 km length transects, averaged over eight different directions) derived from the SRTM30 elevation data set[55]. River density was calculated as the number of river branches within each grid cell[13], derived from the Global Self-consistent, Hierarchical, High-resolution Shoreline database[56].

Vascular plant richness data was from Kreft & Jetz[57]. Species richness of all the amphibian, mammal, and bird taxa was from BiodiversityMapping[58,59] that produces maps of species richness from species distribution data obtained from IUCN, BirdLife International, and NatureServe databases. These maps were resampled to the grid resolutions we used in our analyses. To capture broad scale variation in ecosystem structure and composition, we also compiled data on the world's biomes from WWF[60]. Biomes are discrete regions with a distinct ecological character that is determined by a combination of climate, geomorphology and vegetation types[60].

**Statistical analysis**. We applied generalized least squares (GLS) analysis, implemented in the R package *nlme*[61], to fit regression models to log-transformed language diversity. We did not transform predictors, because residuals in language diversity after accounting for all the untransformed climate and landscape predictors do not violate normality under any resolution according to Shapiro-Wilk normality test (low: $p = 0.72$; medium: $p = 0.19$; high: $p = 0.07$). We accounted for spatial autocorrelation and phylogenetic relatedness by constraining the residual correlation in language diversity between each pair of grid cells to be a linear function of the spatial proximity and phylogenetic similarity between the two cells. The correlation matrix has the form: $(1 - \alpha)I + \alpha[\beta P + (1 - \beta)D]$, where $I$ is an identity matrix, $P$ is the phylogenetic similarity matrix, and $D$ is the spatial proximity matrix, $\alpha$ represents the relative contribution of spatial and phylogenetic versus other residual effects, $\beta$ represents the relative contribution of spatial versus phylogenetic effects[62]. Because our analysis controls for non-independence of grid cells, we can be more confident that the results are not driven by pseudoreplication. For example, without such correction, grid cells in the Arctic that repeatedly sample the same widely distributed languages (e.g., Russian and Yakut) may have a disproportionate influence on global language diversity correlations (Supplementary Figure 1).

In order to correct for non-independence owing to descent, we need a matrix of covariation representing expected patterns of similarity. There is no accepted universal phylogeny for the world's languages, so we constructed a global hierarchy of language relationships from the World Language Mapping System[1] taxonomy using the python library Treemaker[63]. This hierarchy is a proxy for the expected patterns of similarity due to relatedness and does not represent a phylogenetic history of descent. It is a representation of the relationships within language families and therefore provides a way to generate a matrix of expected similarity due to descent[33]. The global language taxonomy is only resolved to the language family level, so we assume that any pair of languages from different families represent the maximum distance from each other. This hierarchy is therefore unresolved at the base. The expected similarity due to relatedness of languages was calculated for each pair of grid cells using the *PhyloSor* metric[64]. This measure compares the sum of distances on the language hierarchy that connect all the languages that occur in a pair of grid cells to the sum of distances that connect all the languages occurring in each grid cell. This measure ranges from 0 to 1, with 0 for two grid cells that do not share any language families in common and 1 for two grid cells that have an identical set of languages.

The spatial proximity matrix was derived from the great-circle distances between the centroids of each pair of grid cells. We modeled the decay in similarity of language diversity with distance as the Gaussian function $e^{-(d/\gamma)^2}$, where $d$ is the great-circle distance between the two grid cells and $\gamma$ is the coefficient describing how fast similarity decays over the distance between grid cells.

Adjacent grid cells can share similar or identical values for environmental variables, as well as sharing many of the same species and languages, making their correlation coefficient at or close to 1. A large number of self-similar values lead to degeneracy of the matrix (with much less information than the number of entries in the matrix). Under medium and high resolutions, correlation between adjacent grid cells is so high that the correlation matrix is nearly singular, leading to a high level of error when taking the inverse of a large matrix. We limit self-similarity across the correlation matrix by subsampling grid cells to avoid adjacent cells with highly similar values. For medium resolution, we avoided sampling adjacent cells by first removing the nine surrounding grid cells, i.e., sampling a grid cell every two rows and columns. This was insufficient to allow convergence of likelihood estimation for the high-resolution grids, so we then removed the 24 surrounding grid cells, i.e., sampling a grid cell every three rows and columns (Supplementary Figure 6). This resulted in 216 grid cells under low resolution, 192 grid cells under medium resolution, and 366 grid cells under high resolution. This subsampling procedure also has the effect of reducing the disparity in number of datapoints at different resolutions. We repeated analyses under high resolution using subsampling that starts with different rows and columns. For example, starting with row 1, we will sample row 4, 7, etc., while starting with row 2, we will sample row 5, 8, etc. In total, there are nine subsampling regimes. Analyses under different subsampling generate qualitatively the same results (Supplementary Table 3 and 4).

We used the *subplex* method in the R package *nloptr*[65] to find the maximum-likelihood estimates for the coefficients in our regression models. To test if a variable is associated with language diversity above its covariation with the other variables, the variable was dropped from the full model that included all the variables, then a likelihood ratio test was used to test if dropping the variable significantly decreased model fit. To assess how much variance in language can be explained by the climatic and landscape variables, we calculated the predicted $R^2$ of the regression model that included all the climatic and landscape variables as predictors. To evaluate the contribution of phylogenetic non-independence and spatial autocorrelation, we refitted the regression model using the method of

ordinary least squares (OLS), which does not account for correlation structure in language diversity among grid cells. Difference in the predicted $R^2$ between the GLS method and the OLS method quantifies the impact of spatial autocorrelation and phylogenetic non-independence to the results.

**Reporting summary**. Further information on research design is available in the Nature Research Reporting Summary linked to this article.

## Data availability
The language data that support the findings of the study are available from World Language Mapping System but restrictions apply to the availability of these data, which were used under license for the current study, and so are not publicly available. These data are however available from the authors upon reasonable request and with permission of World Language Mapping System. The other data are from published and publicly available databases (see text for details). Publications and web links for these datasets are reported in the references. Figures 1, 2, 4, and Supplementary Figure 2–4 have associated raw data.

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

## Acknowledgements

We thank Holger Kreft & Walter Jetz for supplying species richness data for vascular plants.

## Author contributions

S.G., L.B., X.H., M.C. and H.S. conceived the project and designed the study. H.S., M.C. and S.G. collected the data. X.H. designed and conducted the analyses. X.H., L.B., S.G., and M.C. wrote the paper. All authors edited and approved the manuscript.

## Additional information

**Competing interests:** The authors declare no competing interests.

