## [Peer Review File · Nature Communications]

Editorial Note: This manuscript has been previously reviewed at another journal that is not operating a transparent peer review scheme. This document only contains reviewer comments and rebuttal letters for versions considered at Nature Communications .

Reviewers' Comments:

Reviewer #4:

Remarks to the Author:

I am a new reviewer for "The ecological drivers of variation in global language diversity" and I have read both the comments raised by earlier reviewers and the author's responses. Overall, it is my impression that these responses are both appropriate and sufficient (furthermore, I wish to add that in my professional opinion, some of the criticisms leveraged in the first round of review, while well intentioned, seemed unfair and off point). There are many things to like about this manuscript (sample completeness and statistical rigor particularly stand out) and I strongly disagree with the implication that there is nothing new here. In terms of findings, I would say that perhaps the most surprising result is that isolation metrics are not real good predictors of language diversity (I'll come back to that in a minute). Below I offer a few suggestions to consider.

Minor comments:

- Metrics of isolation. Rivers have many different effects on human populations. They are sources of food and water, and they can either connect or isolate human groups. The authors follow earlier studies and used a painfully simplistic river variable (river density) that misses such reality. As such, it is not surprising that they find that "rivers seem to act more as an ecological resource than a barrier to interaction". It would perhaps be more informative to fine tune the "river" variable to something that most closely captures the spirit of isolation hypotheses. For example, rather than counting the density of rivers of any size, it could perhaps be more relevant to explore the effects of the density of rivers with flows greater than X (X of course should be tuned to the best fitting value) or the number of subsections observed within each cell if you assume that rivers of a given size dissect the land and lead to lower connectivity (i.e., to try to explicitly capture the potential for physical separation). Similarly, landscape roughness (BTW. apologies if I missed it but I don't think the authors mention the scale at which they measure the autocorrelation in elevation) may be less informative on physical separation than mean slope at different scales within the cell (which is not the same as altitudinal range). I completely understand that the authors may not wish to undertake any additional analyses but my point here is that I believe they could consider toning down the claim that isolation mechanisms do not work and discuss the fact that our metrics of this phenomenon are unfortunately less than ideal (it seems much easier to capture the spirit of ecological hypotheses in a meaningful way than isolation mechanisms).

- Phylogenetic distance. There is in fact a global phylogeny of human languages (Jäger 2018 Scientific Data 5: 180189) that while not necessarily well received by linguists, would allow more nuanced measurements of phylogenetic distances among the languages in this study. If the authors consider that this is not a suitable resource (and they very well might) they should at least acknowledge it and state why they prefer to stick with the coarser metric they are currently using (right now the text seems to be implying that there is simply no global phylogenetic hypothesis and this is not true).

- Number of speakers. The data on language ranges understandably is restricted to our best guesses of traditional "homeland" ranges (whatever that is considering that even in their early stages, languages probably varied in range!). Similar care does not seem to have been placed on the data on number of speakers. Am I right in understanding that these numbers simply refer to the total number

of speakers for a given language today regardless of where they reside? It honestly seems to me that this is an area where you should accept your limitations and simply say that there are no good data on number of speakers. It seems disingenuous to claim that you have tested the predictions related to this variable when the data themselves are very likely to have nothing to do with what they are supposed to be measuring.

- Statistical Analyses: You say that you report results from models with non-transformed predictors because this does not change the significance of your results. The goal of data transformation is not to make things simpler or easier to understand, it is to make sure that the model assumptions are met. Your statement therefore confuses me: are the residuals of model with untransformed predictors still normally distributed? Are those models appropriate?

- Subsampling: How many replicates of the subsampling process were run? how consistent were the results among them? Unclear

Reviewers' comments:

Reviewer #4 (Remarks to the Author):

I am a new reviewer for "The ecological drivers of variation in global language diversity" and I have read both the comments raised by earlier reviewers and the author's responses. Overall, it is my impression that these responses are both appropriate and sufficient (furthermore, I wish to add that in my professional opinion, some of the criticisms leveraged in the first round of review, while well intentioned, seemed unfair and off point). There are many things to like about this manuscript (sample completeness and statistical rigor particularly stand out) and I strongly disagree with the implication that there is nothing new here. In terms of findings, I would say that perhaps the most surprising result is that isolation metrics are not real good predictors of language diversity (I'll come back to that in a minute). Below I offer a few suggestions to consider.

Minor comments:

- Metrics of isolation. Rivers have many different effects on human populations. They are sources of food and water, and they can either connect or isolate human groups. The authors follow earlier studies and used a painfully simplistic river variable (river density) that misses such reality. As such, it is not surprising that they find that "rivers seem to act more as an ecological resource than a barrier to interaction". It would perhaps be more informative to fine tune the "river" variable to something that most closely captures the spirit of isolation hypotheses. For example, rather than counting the density of rivers of any size, it could perhaps be more relevant to explore the effects of the density of rivers with flows greater than X (X of course should be tuned to the best fitting value) or the number of subsections observed within each cell if you assume that rivers of a given size dissect the land and lead to lower connectivity (i.e., to try to explicitly capture the potential for physical separation). Similarly, landscape roughness (BTW. apologies if I missed it but I don't think the authors mention the scale at which they measure the autocorrelation in elevation) may be less informative on physical separation than mean slope at different scales within the cell (which is not the same as altitudinal range). I completely understand that the authors may not wish to undertake any additional analyses but my point here is that I believe they could consider toning down the claim that isolation mechanisms do not work and discuss the fact that our metrics of this phenomenon are unfortunately less than ideal (it seems much easier to capture the spirit of ecological hypotheses in a meaningful way than isolation mechanisms).

> We agree that rivers are unlikely to represent the same kind of isolating mechanisms for humans as they do for many other species, and indeed this is one of the reasons that the results of the previous study were surprising and merited reanalysis. We deliberately use the same metrics as previous studies as one of our aims is to investigate whether their conclusions are robust to a different analysis. Similarly, we use landscape roughness because this too has been previously correlated with language diversity and we wanted to reexamine that result in light of analyses controlling for non-independence and other issues.

The reviewer is correct that from this we cannot conclude that isolation is unimportant, but that these metrics do not have explanatory power as previously claimed. We have adjusted the text in many places in the manuscript accordingly. For example, we have removed the claim from the abstract that we find no support for the isolation hypothesis. We emphasize that while there is a correlation between river density and altitudinal range and language diversity, the predictions of the isolation hypothesis do not explain these observations, because there is no apparent association between these landscape features and the average size of speaker populations.

P2: we make it clearer that we are testing previously suggested measures of isolation: “Previous studies have suggested that geographic correlates of language diversity such as river density¹³, landscape roughness^{13,14}, elevation range¹⁵, and habitat diversity¹⁴, pointing to a role for isolation in generating language diversity.”

P9: We have change the statement “we find little consistent support for effect of isolation on language diversity” to “we find little consistent support for effect of landscape factors on language diversity.”

P15: We have added the following statement to clarify the measure of roughness: “autocorrelation in altitude¹³ (at every 1km along 100 km length transects, averaged over eight different directions)”.

- Phylogenetic distance. There is in fact a global phylogeny of human languages (Jäger 2018 Scientific Data 5: 180189) that while not necessarily well received by linguists, would allow more nuanced measurements of phylogenetic distances among the languages in this study. If the authors consider that this is not a suitable resource (and they very well might) they should at least acknowledge it and state why they prefer to stick with the coarser metric they are currently using (right now the text seems to be implying that there is simply no global phylogenetic hypothesis and this is not true).

> The Jäger paper was published after we submitted our manuscript, so we could not have used it for our analysis. To rerun the analysis at this stage would take many weeks of computer time, and it would be unlikely to change the results as they use the same source of information on relationships: as stated by Jäger “While these trees still await a detailed qualitative assessment by trained comparative linguists, they are (by construction) compatible with the Glottolog”. However, we now cite this paper in our methods (page 12). Note that the Jäger phylogeny is not a dated phylogeny, so it would not solve the problem we allude to in the passage that refers to the lack of a global phylogeny of languages.

- Number of speakers. The data on language ranges understandably is restricted to our best guesses of traditional "homeland" ranges (whatever that is considering that even in their early stages, languages probably varied in range!). Similar care does not seem to have been placed on the data on number of speakers. Am I right in understanding that these numbers simply refer to the total number of speakers for a given language today regardless of where they reside? It honestly seems to me that this is an area where you should accept your limitations and simply say that there are no good data on number of speakers. It seems disingenuous to claim that you have tested the predictions related to this variable when the data themselves are very likely to have nothing to do with what they are supposed to be measuring.

> We used the WMLS database which is the number of L1 speakers within a given country – we now specify this in the methods. We agree that this is not a perfect measure of speaker population size throughout the history of the language – but such a measure does not exist, because we cannot reliably project current population data backward in time (of course the same problem also applies to nearly all other variables in the dataset, a point acknowledged on page 13). While the data used are imperfect, we believe they are adequate for purpose. We have previously demonstrated that modeling projected change in population over time does not necessarily give a better fit to the data (Bromham et al PNAS 112: 2097, Greenhill et al Frontiers in Psychology, 9:576). The difference between in-area and out-of-area speakers will

not influence the majority of languages in our analysis, as most indigenous languages have few out-of-area speakers. However, we acknowledge the limitation of these data in our methods (page 15).

- Statistical Analyses: You say that you report results from models with non-transformed predictors because this does not change the significance of your results. The goal of data transformation is not to make things simpler or easier to understand, it is to make sure that the model assumptions are met. Your statement therefore confuses me: are the residuals of model with untransformed predictors still normally distributed? Are those models appropriate?

We have now reported the results of normality test on the residuals of the regression model that includes our untransformed climate and landscape predictors. We have rewritten the sentence as:

“We did not transform predictors, because residuals in language diversity after accounting for all the untransformed climate and landscape predictors do not violate normality under any resolution according to Shapiro-Wilk normality test (low: $p=0.72$; medium: $p=0.19$; high: $p=0.07$).”

- Subsampling: How many replicates of the subsampling process were run? how consistent were the results among them? Unclear

We have replicated the subsampling process in order to evaluate consistency across alternative subsampling replicates. We have added the following description to the methods:

“We repeated analyses under high resolution using subsampling that starts with different rows and columns. For example, starting with row 1, we will sample row 4, 7, etc., while starting with row 2, we will sample row 5, 8, etc. In total, there are nine subsampling regimes. Analyses under different subsampling generate qualitatively the same results (Supplementary Table 3-4).”